# Antibiotic Pollution in the Environment: From Microbial Ecology to Public Policy

**DOI:** 10.3390/microorganisms7060180

**Published:** 2019-06-22

**Authors:** Susanne A. Kraemer, Arthi Ramachandran, Gabriel G. Perron

**Affiliations:** 1Department of Biology, Concordia University, 7141 Sherbrooke Street W, Montreal, QC H4B1R6, Canada; a_ramac@live.concordia.ca; 2Department of Biology, Reem-Kayden Center for Sciences and Computation, Bard College, 31 Campus Road, Annandale-On-Hudson, NY 12504, USA; gperron@bard.edu; 3Center for the Study of Land, Water, and Air, Bard College, Annandale-On-Hudson, NY 12504, USA

**Keywords:** antibiotic resistance, environmental resistome, antibiotic pollution, antimicrobial resistance (AMR) policies

## Abstract

The ability to fight bacterial infections with antibiotics has been a longstanding cornerstone of modern medicine. However, wide-spread overuse and misuse of antibiotics has led to unintended consequences, which in turn require large-scale changes of policy for mitigation. In this review, we address two broad classes of corollaries of antibiotics overuse and misuse. Firstly, we discuss the spread of antibiotic resistance from hotspots of resistance evolution to the environment, with special concerns given to potential vectors of resistance transmission. Secondly, we outline the effects of antibiotic pollution independent of resistance evolution on natural microbial populations, as well as invertebrates and vertebrates. We close with an overview of current regional policies tasked with curbing the effects of antibiotics pollution and outline areas in which such policies are still under development.

## 1. Introduction

The discovery of penicillin by Alexander Fleming, in 1929, is often described as one of the most important medical discoveries of the twentieth century [1]. By inhibiting the cell wall biosynthesis of pathogenic bacteria, penicillin was able to stop infectious pathogens such as *Staphylococcus aureus*, a leading cause of death in European hospitals at the time [2]. Between the 1940s and early 1970s, the growing modern pharmaceutical industry, heavily influenced by antibiotic discovery, commercialized over 160 new antibiotics and semi-synthetic derivatives molecules, which consequently became the foundation for the treatment of infectious diseases [3]. Yet, despite great success in reducing mortality and morbidity due to common infections, bacteria able to tolerate or resist the action of antibiotics were quickly observed in laboratories and shortly after in clinical medicine [4,5,6].

Today, the evolution of microbial pathogens able to resist antibiotics treatments is seen as one of the most pressing public health crises [7,8,9,10]. The European Centre for Disease Prevention and Control estimates that each year, 25,000 people in Europe die directly from drug-resistant bacterial infections [11], while recent estimates provided by the British government suggest a worldwide mortality of half a million people [10]. Antibiotic resistance also imposes a significant financial burden on world economies, with the USA alone spending an estimated $35 billion per annum on the treatment of resistant infections [9]. To make matters worse, the rate of antibiotic discovery has declined over the past decades due to technical and economic challenges, leading up to an “antibiotic crisis” [12]. This prognostic directed world leaders to call for an immediate reduction in antibiotic use [8,13,14].

Yet, the global use of antibiotics increased steadily over the past decades, both due to an augmentation of antibiotic use in human medicine and in other sectors of commercial activity [15]. For example, antibiotic consumption in livestock reached 63,151 tons in 2010 and is predicted to increase by another 67% by 2030 [16]. Antibiotic use is also rising in aquaculture, the fastest-growing food sector worldwide due to intensive farming [17]. For this reason, antibiotics of pharmaceutical origin are now found in large quantities in human-made environments such as sewage and waste water treatment plants (WWTPs) [18]. Moreover, because antibiotic pollution is poorly regulated on a local and global scale, antibiotic molecules are increasingly found in terrestrial, freshwater, and marine environments [19].

In this review, we aim to discuss the causes and effects of the presence of antibiotics in the environment, both in terms of the evolution and spread of antibiotic resistance, as well as direct impact of antibiotics as environmental pollutants. In the first part, we outline the origin and flow of antibiotic resistance genes (ARGs) and antibiotic resistance-carrying bacteria (ARBs) in different environments. We then summarize several important vectors of ARGs and ARBs transmission. In the second section, we discuss the possible effects of antibiotic pollution independent of resistance evolution both on endemic microbial communities as well as on higher organisms in different environments. We close this review with a section outlining international policy approaches aiming to mitigate the spread of antibiotics and antibiotic resistance in the environment and highlight gaps in current policies.

## 2. An Overview of Antibiotic Resistance in the Environment

One of the most noted consequences of antibiotic misuse and antibiotic pollution is the increased frequency of bacteria harboring ARGs in different environments (here, antibiotic resistance is defined as any reduction in susceptibility in a bacterial strain compared to the susceptible wildtype [20,21]). While early antibiotic treatments showed great promise in treating bacterial infections, leading some research to proclaim the elimination of infectious diseases, antibiotic-resistant bacteria were quickly observed following the application of antibiotics at a larger scale [5,22,23,24]. In fact, the speed of resistance emergence is remarkable: as an example over 70% of *S. aureus* strains isolated became resistant to erythromycin (first used in 1952 to treat infections that had become resistant to penicillin) within just six months after the onset of treatment [22]. An increase of antibiotic-resistance genes has also been observed in environmental samples. For example, ARG abundance for all classes of antibiotics were found to be significantly increased in soils from the Netherlands since the 1940s [25]. 

Resistance to antibiotics can be conveyed via a broad range of mechanisms [26,27]. For example, antibiotics can be inactivated (e.g., beta-lactamases cleaving beta-lactams such as penicillin) or transported outside of the bacterial cell via efflux pumps (e.g., TetA proteins pumping tetracyclines outside of cells). The modification of the antibiotic’s target (e.g., point mutations in *gyrA* prevent binding by ciprofloxacin) is another common mechanism. 

As a response to the application of antibiotics, resistance can generally arise by two different mechanisms: 1) Resistance can occur via de novo mutation or 2) ARGs may already be present in the environment and increase from rare. While de novo resistance evolution in response to anthropogenic selective pressures has received much attention [28,29], recent work has highlighted that antibiotic production in tandem with antibiotic resistance is wide-spread and ancient in microbial ecosystems. The ecological role of these antibiotics is multi-faceted and not well-understood. While antibiotics may be used as weaponry in competitive interactions in multiple environments ranging from the rhizosphere [30] to the water column [31], their concentrations in the environment may often be too low to effectively kill competing bacteria. Concentration-dependent effects of antibiotics have been described in a hormensis framework [32], which has shifted the focus of research from antagonistic interactions towards low concentration (sub-minimum inhibitory concentration: sub-MIC) effects. At sub-MIC concentrations, antibiotics likely act as signals. There is mounting evidence that antibiotics can change transcription profiles [33], for example of virulence factors during infections, and that antibiotic production itself is intimately tied to quorum sensing (QS) [34], with antibiotics maybe acting as QS signal attenuators [35]. Lastly, antibiotics have been shown to stimulate conjugation [36,37].

Given the complex ecological role that antibiotics likely play in microbial ecosystems, it is not surprising that antibiotic resistance is in fact an ancient phenomenon. Recent work has investigated a range of pristine, non-human impacted environments, from permafrost soils to caves, to determine the human-independent extent of antibiotic resistance. As a rule, resistance genes are readily found in such environments and have informed our understanding of the ‘natural’ resistance state of microbial ecosystems [38,39,40,41,42]. For example, 65% of gram-negative and 70% of gram-positive bacterial isolates from a pristine cave microbiome carried multiple functional antibiotic resistance genes, indicating that multiple resistance is the environmental norm, rather than an exceptional response to high anthropogenic antibiotic pressures [38]. Likewise, resistance genes have been found in 30,000- to 5000-year-old permafrost samples, though a clear increase of the abundance of resistance genes in current samples compared to ancient ones is evident [40,41].

While the evolution of antibiotic resistance pre-dates the human use of antibiotic, the increased frequency of antibiotic-resistant bacteria and antibiotic resistance mechanisms in the environment is now referred to as “antibiotic resistance pollution” [43]. As humans, our concern does not usually cover resistance genes present in environmental bacteria, but instead focuses on instances when environmental resistance genes are transferred horizontally into pathogens, limiting our ability to fight infectious diseases with antibiotics. There are a range of resistance gene transfer mechanisms, discussed in detail elsewhere, including conjugative plasmids [44,45], uptake of naked DNA from the environment (transformation) [46,47], and phage-supported transmission (transduction) [48,49].

From a public health point of view, pathogens carrying resistance genes against multiple classes of antibiotics, often referred to as ‘superbugs’, are of special interest. The most well-known ‘superbugs’ are methicillin-resistant *Staphylococcus aureus* (MRSA), vancomycin-resistant Enterococci (VRE) and extended-spectrum beta-lactamases (ESBL)-carrying *E. coli*. The accumulation of such extensive panels of resistance genes has often been associated with the presence of integrons [50,51]. Integrons are mobile element-located genetic structures which act as natural gene capture systems that can integrate individual gene cassettes such as resistance genes and put them under the control of a strong endogenous promoter [52]. Integrons are so strongly associated with antibiotic resistance that their characteristic *int* gene has been included in many resistance gene panels as a proxy for the presence of ARGs [53].

Importantly, the combination of different resistance genes into a single integron can lead to complex patterns of cross-selection, in which a range of ARGs is maintained due to selection on nearby loci. For example, multi-resistant hospital strains of Enterobacteria were found to carry, among other resistance genes, genes conferring resistance to chloramphenicol, streptomycin, or spectinomycin in their integrons, even though these antibiotics had not been used in the hospital setting for decades [54]. Adding another layer of complexity, integrons often combine ARGs and genes involved in heavy-metal detoxification, which may lead to complex co-selection dynamics between the two [55]. Lastly, cross-selection can occur on the gene level, for example, if a resistance gene is also involved in the transport of other compounds out of the cell [56,57]. 

Anthropogenic usage of antibiotics has essentially started an enormous experimental evolution experiment in nearly every environment on earth. In the following section, we will outline how resistance genes are selected for and flow between different environments, as well as discuss several vectors which contribute to or impact their spread.

### 2.1. Resistance Genes in the Environment

During the last decades, an increasing awareness of the dangers posed by a post-antibiotic era, in conjunction with a wider availability of detection techniques, has led to an exponential increase in work documenting antibiotic resistance in the environment via culture-dependant techniques, conventional PCR, qPCR, or metagenomic methods [58,59,60,61]. The risk of a specific environment being contaminated with ARBs or ARGs is often based on a schematic model of the flow of ARBs and ARGs between different environments, such as the one outlined in Figure 1. Hotspot environments of ARGs and ARBs, where bacteria readily encounter high and repeated doses of antibiotics and have high growth rates due to an abundance of nutrients, have received extended scrutiny [62,63]. In the following section, we will describe several hotspot environments such as hospitals, animal feeding operations (AFOs), aquaculture operations, and wastewater treatment plants (WWTPs), the associated microbiomes under selection, and the flow of ARGs in and out of them.

The prevalence of nosocomial (hospital-acquired) infections with resistant bacteria make hospitals and extended care facilities high interest environments to study the evolution and dissemination of antibiotic resistance. The microbial communities mostly associated with ARGs in hospitals are members of various human microbiomes (e.g., [64]) as well as situated in hospital water and air flow systems [65,66,67]. Hospitals employ a broad range of antibiotics over extended time spans, thus enabling de novo resistance evolution, for example during long-term treatment of chronic infections [68]. Pathogens carrying a newly evolved ARG can subsequently spread between patients epidemically [69] or the gene can be transmitted into other genetic backgrounds via horizontal gene transfer (HGT) [54]. In addition to in-house evolution of resistance, pathogens carrying resistance genes may enter the hospital environment via infected patients, where they can spread epidemically or combine into a new genetic background. The prevalence of different resistance genes encourages their combination into the same genetic background, for example via integrons, resulting in multi-drug resistant pathogens, such as MRSA. Owing to their two-fold role in curing infections with ARBs while at the same time potentially generating them, hospitals have been under intense scrutiny [70]. Hospital effluents have been found to be enriched for antibiotic resistant pathogens such as *E. coli* carrying ESBL, the resistant opportunistic pathogen *P. aeruginosa* and Vancomycin-resistant enterococci (VRE) [70,71]. However, even though their role in generating, concentrating, and disseminating resistance genes seems evident, the overall role of hospitals as sources of environmental resistance is more disputed. This is due to the fact that even though hospitals are enriched for ARBs and ARGs, their contribution to total wastewater is relatively low compared to the general populace (~1% [21]). Estimates of hospital contribution to environmental resistance vary, but may be as high as 33% for specific ARBs ([70] and references therein).

Even though hospitals are under extended scrutiny, they provide relatively controlled environments for the usage of antibiotics and resistance evolution is relatively easy to track (for example via testing of hospital effluent), and subsequently curb. In contrast, antibiotic usage by the general populace is largely unsupervised. The evolution of resistance in human microbiomes, and especially the gut microbiome, via in-home antibiotics use can be a result of inappropriate use of antibiotics or incomplete treatment, resulting in sub-inhibitory antibiotics concentrations in situ. Likewise, even when correctly used, approximately 70% of antibiotics pass the human digestive system unaltered and are excreted via urine [21]. These antibiotics, in conjunction with ARBs and ARGs, are all combined in municipal sewage. The sheer amount of hospital-independently produced sewage, in comparison to hospital-derived sewage, results in the majority of ARGs originating from municipal sewage from the general populace [21].

Sewage from hospitals and the general populace are combined and ultimately transported to WWTPs which employ various biological and physico-chemical processes to biodegrade sewage, reduce the number of pathogens, and remove nitrogen and phosphorous before mixing their discharge with surface water. Sewage constitutes a particularly nutrient rich environment, supporting high concentrations of bacteria, and is steadily seeded with new ARBs, ARGs, and antibiotics themselves. Moreover, a range of antibiotics is present at sub-MIC concentrations, likely encouraging HGT [59,72]. For these reasons, sewage and WWTPs have been described as hotspots for the evolution, recombination, and dissemination of antibiotics resistance [18]. Despite this central role in the antibiotic resistance crisis, WWTPs are currently not specifically tasked with the removal of ARGs [73]. While total pathogen abundance is decreased via passage through a WWTP, strong selection during treatment regime may actually increase the fraction of resistant pathogens [18], and ARGs are readily detected in the effluent of WWTPs [70,71,72]. While ARB and ARG abundances are highly reduced in the water fraction of sewage [71,74,75], WWTP treatment is much less successful in reducing the abundance of resistance genes in biosolids [74,76]. While biosolids have been buried in landfills in the past, there are current policy shifts to utilise them as fertilisers for crops. The possible contamination of soils, crops and grazing livestock with ARGs and resistant human pathogens thus needs to be investigated [77]. Lastly, it is important to point out that different types of WWTPs employ different kinds of sewage treatments that may be able to curb the enrichment of ARGs in effluent and biosolids to different degrees [71,75,78].

Hospitals, sewage, and WWTPs are primarily charged with human-associated ARBs and antibiotics. These hotspots are of special interest for public health, as potentially resistant pathogens and commensals present in these environments do not need to cross a species barrier to cause infections in humans. Moreover, resistance evolution to antibiotics used in human medicine directly impacts our ability to fight infections in the general populace. The resulting prolonged treatment with a broader spectrum of antibiotics can thus contribute to a feedback loop of resistance evolution between these environments and the general population. As antibiotic usage in the general populace is difficult to control, WWTPs may provide a pivotal point of intervention to curb resistance evolution and transmission.

Independent of their application in human health, antibiotics are extensively used in the production of livestock in animal feeding operations (AFOs), and specifically large scale (over 1000 animals) concentrated animal feeding operations (CAFOs), as well as aquacultures. The exact amount of antibiotics used in such facilities is difficult to estimate, but it is much higher than the amount used by hospitals [79]. Approximately 80% of antibiotics sold in the U.S. are for veterinary use [80] and the concentration of antibiotics used by biomass in animals in Canada is at least twice that used in humans [81]. Antibiotics in AFOs are not only used to treat acute infections but also pre-emptively on the herd level and to promote growth [79,82], leading to intensive selection on microbiomes associated with farm animals and resulting in manure and wastewater contaminated with ARBs [83].

The waste systems used by AFOs are mostly independent of municipal sewage and WWTPs. Manure from AFOs is often collected in waste lagoons where it degrades [79]. Storage of manure reduces the total amount of culturable bacteria, but nonetheless lagoons and manure have been shown to be highly enriched for ARBs [60,61,83,84]. Subsequently, manure is often used as fertiliser on nearby crop lands, where ARGs and surviving ARBs come into contact with the soil microbiome. Via run-off, ARGs and ARBs from manure can reach both surface and ground water [79].

While we have so far focussed on animal production on land (e.g., poultry, pigs, and cows), aquaculture installations, in which fish are farmed, are another livestock production environment where large amounts of antibiotics are regularly employed [85]. In open-water aquaculture, antibiotics are directly added to pens to prevent disease and promote growth, leading to the evolution and widespread dispersal of ARBs and ARGs in sediments and open waters [86,87]. Of special interest are closed systems, in which human or animal waste is fed to fish in aquaculture, as they may contribute to the transfer of resistance genes between systems [88].

Antibiotic resistance caused by the environmental release of antibiotics from industrial producers has received less attention than the somewhat controlled release from AFOs and hospitals. However, the sheer magnitude of contamination with antibiotics can be staggering. For example, effluent from a WWTP in Hyderabad, India, downstream of 90 antibiotics production facilities was found to be highly enriched for clinically important drugs such as ciprofloxacin, leading to concentrations one million times higher than average waste water concentrations for the drug [89,90]. Unsurprisingly, such high contaminations of antibiotics have resulted in increased antibiotic resistance in environments impacted by industrial-scale antibiotics contamination [91,92].

### 2.2. Vectors of ARB Transmission

The transport of ARBs and ARGs from hotspots of resistance evolution to pristine environments occurs via a range of different vectors (Figure 1) (here, we use the term ‘vector’ to describe a mode of transport of ARBs or ARGs between different environments, rather than in the narrower sense of a ‘genetic vector’ shuttling ARGs between different bacterial populations). Vectors may be either purely involved in transport or offer a reservoir in which ARBs and ARGs persist, multiply, or evolve. While ARGs are likely to be transported between environments within ARBs, most studies do not investigate this directly, but instead focus on testing different potential vectors for the presence of a panel of resistance genes. Consequently, the genetic context of most ARGs in the environment is unknown.

#### 2.2.1. Surface Waters

Surface waters receive WWTP and pharmaceutical effluent as well as run-off from manure-fertilized fields and AFOs and are thus located at a central hub for the transport and dissemination of ARBs [61,93,94,95]. ARG profiles found in rivers were strongly impacted by the contamination source (e.g., WWTP or AFO), indicating that ARBs are transported rather than selected there. In conjunction with the presence of ARGs in rivers, high levels of gene transfer elements to facilitate HGT were detected as well, indicating the potential for ARGs to change their genetic background during transfer [96].

#### 2.2.2. Air

Recently, the potential of ARB dispersal via air has received much attention. Indeed, ARGs against seven commonly used antibiotics were readily detected in the particulate matter from city air worldwide, and their relative abundances were highly variable [97]. Likewise, particulate matter from smog was enriched for ARGs [98]. It is interesting to speculate where these ARGs originate. ARGs are most likely associated with the rich diversity of bacteria attached to particles floating in the air and disperse via wind, fog, and precipitation [99]. In fact, different Pseudomonas species including the opportunistic pathogens *P. aeruginosa* and *P. synringae*, which often harbor multiple drug resistance [100,101] have been associated with precipitation [102]. ARGs are also well documented in close proximity and downwind of WWTPs [103], where they may pose a specifically high risk for workers. AFOs and specifically CAFOs have also been shown to seed the air with ARGs [104,105] Lastly, airborne transmission of ARBs from hospitals is less studied, but airborne ARGs have been found there [106]. While seeding of the air with ARBs has been frequently demonstrated, it is difficult to infer how and to which degree relatively local events like the production of an aerosol in a WWTP influence the overall loading of air with ARBs.

#### 2.2.3. Animal Vectors

In contrast to surface waters and air, animal vectors, and specifically their microbiomes, offer nutrient-rich environments that can be colonized by ARBs. Thus, ARGs may amplify or change genetic background within animal vectors, which may constitute a reservoir of antibiotic resistance. Humans are the most studied animal vector (e.g., [107,108]). Farmers and other workers in direct contact with livestock have been shown to develop increased resistance [109,110,111,112]. ARBs from animals have to cross a species barrier to establish themselves in humans, leading to relatively low risk assessments for this ‘direct contact’ transmission route [113]. However, as most research of animal to human transmission focuses on food-borne infections, the prevalence of animal-originated ARGs transferring into human commensals or pathogens within farm worker microbiomes has to our knowledge not been assessed.

While AFO- and CAFO-originating ARBs have to cross a species barrier, transmission of human-evolved ARBs, for example from hospitals and sewage, should be easily possible. As a consequence, humans in those settings may readily become vectors of ARB transmission and dissemination. Indeed, increased antibiotic resistance and thus a potentially increased ARB transmission risk have been associated with hospital workers [114,115]. Importantly, carriage of ARBs extended to family members of care workers, indicating the potential for a chain of transmission [116]. Even though the presence of ARBs and ARGs in WWTPs, as well as the generation of aerosols carrying ARGs is well documented [18,103], we are not aware of any studies investigating the resistome of WWTP workers.

In addition to health care, farm, and WWTP workers, the dissemination of ARBs has been connected to international travellers. Several studies have demonstrated that commensals and pathogens carrying ARGs can establish themselves in the travellers’ microbiome, and that these ARBs can persist for a length of time after return to the country of origin [107,108]. For example, Swedish students undergoing an exchange to India were significantly more likely to return home carrying *E. coli* with ESBL genes than those staying in Central Africa, and establishment of ARB strains in the students’ microbiomes occurred without any antibiotic treatment [107]. The conditions for a successful colonization of a healthy microbiome with environmental ARBs thus warrant further investigation.

Apart from humans, other animals have been described as ARB-carrying vectors. Insects, and specifically houseflies and cockroaches, have been widely associated with transmissions of ARBs from AFOs and CAFOs [117], WWTPs [118], as well as hospitals [119,120]. In this case, jumping across the species barrier does not seem to represent a problem for ARBs, and both active proliferation of ARBs [121], as well as horizontal gene transfer of ARGs in the fly gut [122] have been demonstrated.

A range of vertebrates have been shown to carry ARBs [123], but a direct link between resistance gene carriage and a hotspot of resistance evolution (e.g., hospitals, WWTPs, and AFOs) has been rarely demonstrated. However, small rodents trapped on or close to swine farms carry higher loads of ARBs than those trapped in other areas [124,125]. While rodents have been associated with the spread of antibiotic resistance from AFOs, birds, and especially water feeding birds, have been implicated in the spread of resistance from WWTPs. A recent metagenomics study found that birds exposed to waste water are characterized by a higher abundance and diversity of ARGs, and that especially ducks are prolific vectors of resistance transmission due to their feeding habits and habitats [126]. Another group of birds frequently studied in the context of ARB carriage is seagulls [127], which have been shown to carry resistant pathogens such as *E. coli* carrying ESBL [128], presumably contracted via contact with contaminated water [126].

While hotspots for the evolution and dissemination of ARBs such as hospitals AFOs and WWTPs have been thoroughly assessed, the importance and risk posed by vectors of transmission between environments has not been systematically studied. As vectors may aid in the multiplication of ARBs and recombination of ARGs and offer unique selective conditions, they are central to understanding the flow of antibiotic resistance between environments. Specifically, the risks of combining resistance genes from human and animal systems and the crossing of the species barrier need to be assessed. Ultimately, ARBs are deposited into the environment via vector transmission. Their possible fates there are outlined in Box 1.

Box 1What is the ultimate fate of ARBs and ARGs in environments?ARBs:-Due to their adaptation to the gastrointestinal tract, fitness of many ARBs in soil and water environments is expected to be low, leading to persistence times in the order of weeks to months [129,130,131].-Resistance-gene carrying opportunistic pathogens or commensals from hospital and AFO environments such as *Acinetobacter*, *Pseudomonas*, or *Aeromonas* may have wider niche breaths and may thus be equipped to survive longer or even thrive in soil and water [132].-ARBs could lose ARGs when the selective pressure of the antibiotic is no longer present (the cost of resistance hypothesis [133]), but this is rarely supported by experimental data, likely due to the evolution of compensatory mutations and low-cost resistance mutations causing AR [134].ARGs:-The persistence of resistance genes in the soil environment is variable and depends on a range of environmental factors including temperature, moisture, pH, and the microbial community present [135,136].-The transformation of naturally competent soil bacteria such as *Acinetobacter sp.*, *Burkholderia sp.*, and *Pseudomonas fluorescens* with ARGs is possible [137,138]. -Human pathogens and soil bacteria have been shown to have overlapping resistomes, indicating a vast potential for HGT between the two groups [139].

## 3. Cytotoxic Effects of Antibiotics in the Environment

Even though research often focusses on antibiotics as means to treat or prevent bacterial infections, most clinical antibiotics are derived from biomolecules naturally produced by soil-dwelling microorganisms as secondary metabolites [140]. As mentioned above, the native roles of antibiotics are multi-facetted and include pigments, toxins, and effectors of various kinds [141]. For this reason, it is still a matter of active debate whether antibiotics evolved to harbor inhibitory activity under natural condition [142,143]. However, resistance genes such as β-lactamases are predicted to have originated millions of years ago [141,144,145], suggesting that antibiotics have been modulating microbial populations long before their use in clinical medicine. Nevertheless, the amounts of antibiotics released and accumulating in the environment resulting from human activity are unprecedented, with potential impact on local communities of microorganisms and animals alike. In this section, we will focus on the direct, resistance evolution-independent, effects of antibiotic pollution.

### 3.1. Antibiotic Pollution Disrupts Microbial Communities

#### 3.1.1. Microbial Evolution

In addition to favoring the evolution and spread of antibiotic resistance as discussed above, the selective pressures imposed by antibiotic pollution can impact on the evolutionary dynamics observed in microbial populations in different ways [146]. Different bacterial species or even lineages within a species harbor phenotypic heterogeneity in their response to antibiotics and environmental stresses more broadly. For example, certain bacterial populations present different levels of tolerance to antibiotics due to changes in gene expression or fluctuations in critical physiological traits [147,148]. On the one hand, antibiotics can thus reduce diversity in microbial populations, by favoring the growth of resistant or tolerant microbial lineages under strong selective pressures. On the other hand, weak selective pressures imposed by lower antibiotic concentrations can selectively favor the growth of bacterial lineages with increased phenotypic and genotypic diversity. Indeed, intermediate concentrations of antibiotics such as amikacin, ciprofloxacin, and streptomycin were found to select for larger colony size in *Staphylococcus* spp. [149], a trait which was linked to increased genetic diversity and adaptability in several species of bacteria [150,151]. 

In addition to changes in the population composition of bacteria, exposure to low levels of antibiotics has also been shown to increase genetic diversity in microbial population via the activation of the bacterial SOS response, resulting in an increased mutation rate throughout the genome [152], and via the direct mutagenic effect on the DNA [153]. Antibiotics were also found to increase the horizontal transfer of genetic material between bacteria either by conjugation [154] or by increasing competence resulting in the uptake of extracellular DNA [155]. Lastly, most antibiotics were found to affect gene regulation at the transcription level [33,156], either via direct binding or through other regulatory mechanisms such as riboswitches [157] and quorum sensing [158], resulting in increased phenotypic variability and, in some instances, increased virulence [158]. Together, these mechanisms increase the available pool of genetic and phenotypic diversity in bacterial populations exposed to antibiotics. In turn, this may facilitate the further evolution of antibiotic resistance if selective pressure increases, as predicted by Fisher’s Fundamental theorem of evolution by natural selection [149,159].

#### 3.1.2. Microbial Diversity and Ecosystems Functions

Bacteria and fungi inhabiting soil and water environments are not only the foundation of the most diverse and densely populated ecosystems on Earth [160,161], but are also crucial for the performance of important ecological functions such as nutrient cycling, decomposition, and primary productivity in a range of environments [162,163,164]. Selective pressures associated with antibiotic pollution can act on the overall microbial community composition by reducing taxa diversity or by shifting microbial composition. Generally speaking, antibiotic exposure tends to favour an increase in Gram-negative bacteria as opposed to Gram-positive bacteria. The latter showing increased susceptibility to antibiotics and disinfectants due to the absence of an outer cell membrane [165]. Exposure to antibiotics may thus result in the loss of key microbial taxa filling critical ecological roles.

For example, antibiotic pollution in aquatic environments was found to reduce overall microbial diversity, including taxa responsible for carbon cycling and primary productivity [146,166,167,168]. Similarly, the presence of antibiotics in soil was found to alter microbial community structure, leading to a loss of biomass and a reduction in microbial activity including nitrification, denitrification, and respiration [169,170,171]. Moreover, antibiotics can also affect bacterial enzyme activity, including dehydrogenases, phosphatases, and ureases, which are considered important indicators of soil activity [171]. Finally, antibiotic disruption of microbial communities can also lead to an increased abundance of parasites and pathogens in both soil and water environments. For example, the presence of antibiotic pollution in aquatic environment was shown to lead to an increase frequency of toxic *Cyanobacteria* species, causing eutrophication in freshwater environments and posing health risks to humans [172].

### 3.2. Antibiotic Pollution and Toxicity in Higher Organisms

#### 3.2.1. Physiological Effects

In addition to their impact on microbial populations, antibiotics can also affect higher organisms. Side effects associated with different antibiotics are well documented in human medicine thanks to standardized clinical trials and pharmacology reports [173]. On the other hand, it is sometimes argued that the levels of antibiotic compounds present in the environment are very low and present negligible risks to humans [174]. However, antibiotics present in the environment at low concentrations can accumulate in human populations via long-term exposure to drinking water, food, or consumer goods with unknown health consequences. For example, macrolides and quinolones have been previously detected in chlorinated drinking water [175]. Triclosan, an antimicrobial compound used in consumer goods such as soap and clothes, has been increasingly detected in stream and rivers around the world [176] and, more recently, in human serum, urine, and breast milk of people not currently using the antimicrobial [177], with possible health effects ranging from reproductive problems to muscle weaknesses [178]. In fact, it has been estimated that 75% of the U.S. population has been exposed to triclosan or other antibiotics via consumer goods [179].

While toxicity levels for humans are often a known quantity, there is comparatively little information regarding toxic concentration in wild animals, especially smaller and more susceptible organisms [180,181]. Indeed, low concentrations of common antibiotics such as streptomycin and erythromycin have been shown to impact the survival and behaviours of micro invertebrates such as *Daphnia magna* [182] and *Artemia* in laboratory conditions [183]. Further studies revealed that antibiotic toxicity can be accentuated following exposure to UV radiation, as experienced by most organisms in the wild [184], thus highlighting the difficulty of assessing the risk associated with antibiotic pollution in natural environments through lab studies.

Antibiotic pollution has also been observed to have negative effects on vertebrates. The effect of antibiotic pollution is thought to be especially strong in aquatic environment where animals are chronically exposed to the pollutants [185]. For example, sub-inhibitory concentrations of macrolides were shown to induce malformations in zebrafish such as yolk sac edema and uninflated swim bladder as well as influencing embryo spontaneous movement frequency [186]. Similar results were found in other experimental fish models exposed to antibiotics including sulphonamide [187], quinolone [188], and tetracycline [189]. In addition, quinolones and their metabolites often persisted in the body for a long time [190], contributing to chronic toxicity and the risk of bioaccumulation [185]. While less commonly investigated, antibiotic pollution was also found to be toxic to amphibians, where tetracycline was shown to induce shortened body length, pericardial edema, and other malformation in *Xenopus tropicalis* [185]. 

In addition to physiological effects, antibiotics can also interfere with development and behaviour, potentially resulting from changes in gene expression [191]. In fact, it is now known that most antibiotics, regardless of their receptors and modes of actions, provoke considerable transcription activation at low concentrations in bacteria and multicellular organisms alike [33,156]. Kim and colleagues found that tetracycline exposure affected gene regulation in *D. magna*, mostly impacting the general stress response, as well as protein and carbohydrate metabolism [192]. Moreover, such changes in gene transcription level can carry-over over multiple generations even in the absence of tetracycline [192], thus potentially impacting on animal population even after the removal of the molecule from the environment.

#### 3.2.2. Effect on Host Microbiomes

Antibiotics have also been shown to impact higher organisms by disrupting the microbial populations associated with animal hosts [193]. Animal microbiomes fulfill different functional roles for the hosts, ranging from efficient nutrient metabolism to promoting bone formation [194,195]. Therefore, the disruption of a host’s microbiome, known as dysbiosis, can lead to important health consequences, including developmental defects, allergies, metabolic diseases or increased susceptibility to pathogens [196,197,198,199,200]. 

While the impact of antibiotic exposure on host microbiomes has been mostly studied in human or in the murine model at high concentration [201,202], recent studies suggest that lower concentrations of antibiotics such as those found in aquatic environments could also affect local organisms. Indeed, the fish microbiome, which is especially susceptible to environmental variation [203,204], is thus also highly susceptible to the effect of antibiotics. Low concentrations of antibiotics administered as a prophylactic in aquaculture can reduce microbial diversity in the fishes’ gut [205] and increase mortality [206]. Increased mortality was also observed in fish exposed to WWTP effluent waters contaminated with a mixture of chemicals, including high concentrations of antibiotics [207]. While work done on experimental models suggests that adult fish can recover from short-term exposure to antibiotic [208], recent work demonstrated that concentrations of streptomycin and tetracycline as low as 1 μg/mL could cause dysbiosis and increase mortality in more susceptible zebrafish embryos [189,209]. Other negative impacts of antibiotic exposure on animal health putatively due to the effect of antibiotics on the host microbiome include disruption of the growth cycle, and reduction in fertility [210]. While the full extent of the impact of antibiotic exposure on aquatic organisms remains to be investigated, antibiotic pollution is only adding to the current threats already facing aquatic organisms around the world due to anthropogenic activity [211]. 

## 4. Policy Approaches to Tackle Antibiotic Pollution and ABR

### 4.1. Global Context

The sheer magnitude of the antibiotic pollution and resistance crisis, with the estimated deaths due to antibiotic resistance being 700,000 per year globally [212], makes managing its social and economic fallout a global problem. To address this multi-sectoral issue, it is often framed within “The One Health model”, which links the human, animal, and environment health domains. In order to tackle the antibiotic pollution and resistance crisis on a global scale, the World Health Assembly implemented a global action plan in 2015, thus providing the framework for individual national action plans to develop policies and regulations to combat ABR and antibiotic pollution in general. In the following section, we will give an overview of existing national action plans and policies in connection with known hotspots of antibiotic pollution and ABR evolution. Due to space constrains and readily available information, we will focus on and compare national action plans targeting select regions and representative countries within them: The Pan-American region (case study: Canada), the South-East Asian region (case study India), and Europe.

Policies addressing antibiotic pollution and Antimicrobial Resistance (AMR) have been created in the Pan-American region addressing the misuse of antibiotics in the health and agriculture sectors, but much less attention has been devoted to other sources of exposure. Canada engaged in global efforts to fight antibiotic pollution and ABR and committed multi-sectoral support to the implementation of the WHO Global Action Plan on ABR with its own domestic plan “Tackling Antimicrobial Resistance and Antimicrobial Use: A Pan-Canadian Framework for Action” focusing on three main topics: (1) Surveillance, (2) Stewardship, and (3) Innovation [213].

Inappropriate and rampant use of antibiotics is a major contributor to antibiotic resistance in the South-East Asian region, but comprehensive monitoring systems are absent [214]. The incorrect use of antibiotics can be attributed to relatively lax policies and a lack of awareness, high rates of self-medication due to access to antibiotics without prescription, as well as a lack of education about the correct usage of antibiotics [215]. Inappropriate prescribing is also a contributing factor to increased ABR in the region due to diagnostic uncertainties, incentives from pharmaceutical companies, and patient demand [215]. In addition, this region comprises the main countries housing antibiotic manufacturing sites, providing another large contributing component to antibiotic pollution [89]. In response, India has recently started monitoring and developing antibiotic pollution and ABR policies. For example, India’s medical societies adopted a set of national recommendations in 2012 to promote antibiotic stewardship and the major recommendations were incorporated into the 2017 National Action Plan on Antimicrobial Resistance, based on the WHO Global Plan [216].

Europe has been at the forefront of tackling the antibiotic pollution and antibiotic resistance crisis and was quick to recognize the need to invest in research and policy starting in the early 2000s and implemented the “EU One Health Action Plan against AMR” in June 2017. The key objectives of this plan are: (1) making the EU a best practice region, (2) boosting research, development, and innovation, and (3) shaping the global agenda.

In the following, we discuss antibiotic pollution and AMR policies by the main sources of contamination outlined above, which are agriculture, aquaculture, human medicine, wastewater treatment, and pharmaceutical industry (summarized in Table 1). We will be presenting the policies currently in place that address these key areas as well as outline gaps in existing policies.

Canada, India, and European member states have established and strengthened surveillance systems to identify new threats or changing patterns in antibiotics use and ABR in agriculture and specifically animal production settings, now focusing on the promotion of the appropriate use of antibiotics in veterinary medicine. In this context, the EU has completely banned the use of antibiotics as growth promoters in livestock and food animals since 2006. In Canada, effective as of February 2018, a stronger regulatory framework on veterinary medicines and medicated feeds, including facilitating access to alternatives and encouraging the adoption of better practices in order to reduce the use of antibiotics has been implemented. For example, farmers in Canada are now required to have veterinary prescriptions for antibiotics and medicated feed and growth promotion claims have been removed and responsible use statements have been added to labels of veterinary antibiotics. India is in the top five antibiotic-consuming nations in the food animal sector [214], even though ABR in livestock and food animals has been poorly documented due to few regulations against the use of antibiotics for non-therapeutic purposes [217]. While the use of antibiotics as growth promoters is common practice, there are currently little to no regulations in this sector.

Consistently across Canada, India, and Europe, antibiotics in aquaculture can only be used when they are required to fight disease, and not to stimulate growth. As many pathogens affecting farmed fish can now be prevented using vaccines, quantities of antibiotics used in aquaculture should thus be drastically reduced in these countries, in turn reducing the risk of environmental transmission of ARBs and ARGs [218]. In addition, the small number of antibiotics used to treat farmed fish must be prescribed by veterinarians in both Canada and the EU. However, despite regulatory frameworks, antibiotic misuse is still prevalent in the Indian aquaculture industry, and multi-drug resistant bacteria could be isolated from over two-thirds of aquaculture samples [217].

Policies to increase the awareness of ABR and the risks of over-prescriptions of antibiotics in human medicine were implemented to urge stewardship in patients and healthcare professionals in all three regions, encouraging better practices in human health by avoiding unnecessary use and prescription of antibiotics to cure illnesses. For example, Health Canada is in the process of requesting that drug sponsors update their product labelling with a specific focus on when and how to use antibiotics, thus promoting the choice of the correct antimicrobials for each treatment. 

There are no procedures currently in place to regulate the suppliers in the pharmaceutical industry to ensure that antibiotics are not released into the surrounding waterways during production, with an emphasis on the need for more evidence on the impact of industrial pollution by pharmaceutical companies outlined in the EU action plan [219] and no mention of the topic in the Canadian National Action Plan. The National Action Plan of India [217] is one of the first plans globally, that expresses the intent to create policies that regulate antibiotic residues in industrial effluents, likely due to the country’s role as one of the main manufacturers of antibiotics. 

### 4.2. Gaps in Current Policies

Current research is aimed at investigating the health risks associated with antibiotics and antibiotic resistance genes in environmental reservoirs (e.g., [220]) and the UN has released a report [221] emphasizing the need to seriously consider the risks posed by environmental reservoirs. However, most action plans and policies fail to address the issue, and do not specifically aim to curb antibiotic and ARG pollution of natural environments. 

Likewise, as mentioned briefly above, current policies preventing the environmental release of antibiotics by drug manufacturing facilities are missing in all national action plans. Given the magnitude of localized antibiotic pollution by manufacturers, this omission may facilitate regional hotspots of resistance evolution from which ARBs and ARGs may subsequently spread world-wide.

Lastly, currently no policies are in place in the three regions specifically addressing the problems of antibiotic and ARG pollution of WWTPs. As current WWTP design does not specifically consider the removal of antibiotics, ARBs and ARGs, policies need to be aimed at amending WWTP technologies establishing acceptable ARG load guidelines in WWTP effluents and biosolids. 

## 5. Conclusions

Large-scale antibiotic pollution and the resulting antibiotic resistance is a current major public health problem. While the environmental component of this issue (e.g., environmental reservoirs of resistance genes and the likelihood of horizontal transfer of ARGs between pathogenic and non-pathogenic bacteria) has received an increased research interest in recent years, many dimensions of environmental antibiotic pollution and resistance are still unknown and require further research.

For example, current research of environmental dimensions of antibiotic resistance often tallies up resistance genes found within an environment but gives little indication of the associated risks of transmission, and thus the potential impact on human health. Technically, it is often difficult to infer absolute abundances of antibiotic-resistant pathogens from sequence data, which records relative abundances of resistance genes. Such technical issues aside, only little information exists linking the presence and abundance of ARGs directly to a human’s risk of becoming infected with an antibiotic resistant bacterium upon exposure [222]. One notable exception is a detailed study conducted by Leonard and colleagues [220], in which they calculated the specific risks of ingesting antibiotic resistant bacteria upon engaging in different recreational activities in the contaminated environment. The authors found that despite the low overall prevalence of resistance-gene carrying *E. coli* in coastal waters (~0.12%), certain activities, such as surfing, provided enough contact with the contaminated water to allow transmission. More of such benchmark studies are needed to accurately interpret the environmental resistome data collected and translate it into health risks. 

In addition to contributing to the prevalence of antibiotic resistance, antibiotic pollution also has the potential to affect human and ecosystem health directly. On one hand, the effects of antibiotic pollution are expected to be especially disruptive in aquatic environments, where they can inhibit ecosystem functions and impact on organisms that are exposed throughout their life cycle. On the other hand, the presence of antibiotics in the environment and in animals could also impact on human health. While the effect of such exposure is unknown in humans, epidemiological surveys suggest that long-term exposure to antibiotics may lead to chronic conditions including obesity, diabetes, and asthma [202]. For these reasons, it is imperative to consider the overall impact of antibiotic pollution on humans and on the environment, in addition to its contribution to antibiotic resistance.

Moving from data gathering to more interpretative studies will aid in developing efficient policies born out of the research. Current policy schemes are focussed on surveillance, but the scope of the crisis must eventually lead to forward-thinking policies. Understanding antibiotic pollution and antibiotic resistance as a “One Health Approach” may aid in creating more societal engagement and ultimately more efficient policies. Such policies need to evaluate direct risks of transmission posed by certain contaminated environments while at the same taking into account the complex patterns of inter-environmental transmissions that may exist. As antibiotic resistance and pollution is a global problem and especially rampant in the developing world, international cooperation, data-sharing and globally consistent policies are needed.

## Figures and Tables

**Figure 1 microorganisms-07-00180-f001:**
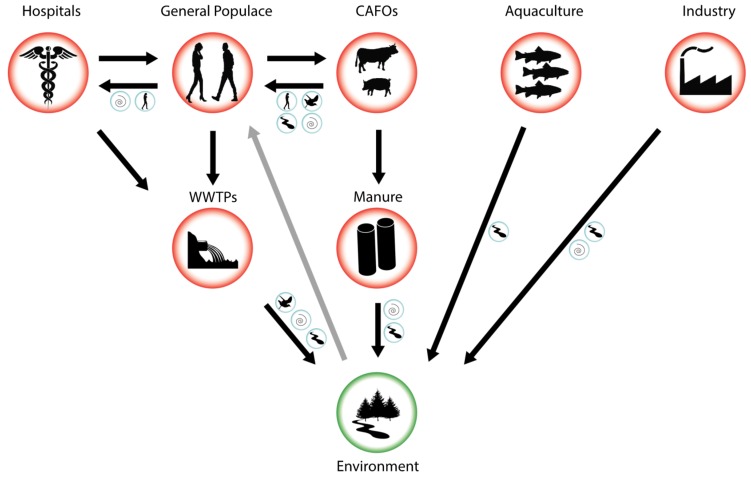
Schematic flow of antibiotic resistance-carrying bacteria (ARBs) and antibiotic resistance genes (ARGs) from hotspots of evolution and transmission (red circles) to the environment (green circle). Blue circles indicate possible vectors that may aid transmission between specific environments including air, surface waters, humans, and other animal vectors. Black arrows indicate known flows of ARBs and ARGs, grey arrow indicates a possible transmission route from a contaminated environment back to the general populace.

**Table 1 microorganisms-07-00180-t001:** Summary of policies in place to address antibiotic pollution and related risks.

	Human Medicine	Agriculture/Livestock	Aquaculture	Wastewater Treatment	Pharmaceutical Manufacturing
Canada	+	+	+	-	-
India	+	-	+	-	-
Europe	+	+	+	-	-

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
