# Peer review of "Antibiotic Pollution in the Environment: From Microbial Ecology to Public Policy"

_microorganisms, 2019, doi:10.3390/microorganisms7060180_

Round 1
Reviewer 1 Report
Overall the review of Kraemer et al., is well documented and written, but I have to say that the explanation of transmission vectors is a bit confusing because authors make no distintion between ARB and ARG. I would say that water, air and animals are vectors of ARB transmission, and vectors for ARGs transmission are bacteria and their mobile genetic elements (which are barely mentioned in the review).
Line 64-65: ARGs and ARBs are already definced in the previous paragraph
Line 66:close parentheses
Line 68: remove one 'quickly'
Author Response
Reviewer 1
Overall the review of Kraemer et al., is well documented and written, but I have to say that the explanation of transmission vectors is a bit confusing because authors make no distinction between ARB and ARG. I would say that water, air and animals are vectors of ARB transmission, and vectors for ARGs transmission are bacteria and their mobile genetic elements (which are barely mentioned in the review).
We thank the reviewer for their comments. We have edited the section in question to make it more precise in accordance with the reviewer’s comment. Firstly, we now explicitly state our definition of the term ‘vector’ to clearly differentiate it from ‘genetic vector’ (which are beyond the scope of this review) (line 263ff.). Secondly, we now explicitly state that while most ARGs are likely situated within ARBs, their exact genetic context is rarely known or even investigated (line 267ff.).
We agree with the reviewer that the section is potentially confusing as we switch between describing ARBs and ARGs associated with vectors. We have clarified the writing throughout the section to avoid confusion. However, in order to not misrepresent the studies that we discuss in this section, we decided to follow the original authors’ leads with regards to whether they investigated ARBs or ARGs independent of their genetic context and reflect the authors’ nomenclature throughout.
Line 64-65: ARGs and ARBs are already defined in the previous paragraph
We removed the definitions in question (line 58ff.)
Line 66: close parentheses (line 61)
Done
Line 68: remove one 'quickly'
Done (line 63)
Reviewer 2 Report
The manuscript prepared by Kraemer is of a very good quality and it deals with an interesting and up-to-date topic, i.e. environmental dimension of antibiotic resistance. In some points this is a typical review article about this phenomenon, but in many fragments I found it novel and refreshing. This is a complex review, thus I strongly recommend its acceptance, after small amendments (please see below).
Minor comments:
1. Line 68. Change “(…) antibiotic‐resistant bacteria quickly were quickly observed (…)” into “(…) antibiotic‐resistant bacteria were quickly observed (…)”.
2. Line 141. Please add “conventional PCR” to techniques mentioned here, as this is still a common technique of ARGs detection.
3. Line 165. Please define an abbreviation “VRE”.
4. Line 183. Please change “the general populace” into “other municipal wastes”.
5. Line 265. Please change “Pseudomonad” into “Pseudomonas”.
Author Response
Reviewer 2:
The manuscript prepared by Kraemer is of a very good quality and it deals with an interesting and up-to-date topic, i.e. environmental dimension of antibiotic resistance. In some points this is a typical review article about this phenomenon, but in many fragments I found it novel and refreshing. This is a complex review, thus I strongly recommend its acceptance, after small amendments (please see below).
We thank the reviewer for their comments and have edited the manuscript according to their suggestions.
Minor comments:
1. Line 68. Change “(…) antibiotic‐resistant bacteria quickly were quickly observed (…)” into “(…) antibiotic‐resistant bacteria were quickly observed (…)”.
Done (line 63)
2. Line 141. Please add “conventional PCR” to techniques mentioned here, as this is still a common technique of ARGs detection.
Done (line 146)
3. Line 165. Please define an abbreviation “VRE”.
Done (line 171)
4. Line 183. Please change “the general populace” into “other municipal wastes”.
Changed into ‘municipal waste from the general populace’ (line 188)
5. Line 265. Please change “Pseudomonad” into “Pseudomonas”.
Done (line 287)